# Procalcitonin Levels in COVID-19 Patients Are Strongly Associated with Mortality and ICU Acceptance in an Underserved, Inner City Population

**DOI:** 10.3390/medicina57101070

**Published:** 2021-10-07

**Authors:** Theresa Feng, Alecia James, Kyra Doumlele, Seth White, Wendy Twardzik, Kanza Zahid, Zeeshan Sattar, Osato Ukponmwan, Mohamd Nakeshbandi, Lillian Chow, Robert Foronjy

**Affiliations:** 1Department of Anesthesiology, State University of New York Downstate Health Sciences University, Brooklyn, NY 11203, USA; theresa.feng@downstate.edu (T.F.); seth.white@downstate.edu (S.W.); 2School of Public Health, State University of New York Downstate Health Sciences University, Brooklyn, NY 11203, USA; alecia.james@downstate.edu; 3Department of Medicine, State University of New York Downstate Health Sciences University, Brooklyn, NY 11203, USA; kyra.doumlele@downstate.edu (K.D.); wendy.twardzik@downstate.edu (W.T.); kanza.zahid@downstate.edu (K.Z.); Zeeshan.sattar@downstate.edu (Z.S.); osato.ukponmwan@downstate.edu (O.U.); Mohamed.Nakeshbandi@downstate.edu (M.N.); lillian.chow@downstate.edu (L.C.)

**Keywords:** biomarkers, ARDS, mortality, COVID-19, intensivist

## Abstract

*Background and Objectives*: This study aimed to identify demographic and clinical factors at the time of critical care consultation associated with mortality or intensive care unit acceptance in a predominantly Afro-Caribbean population during the first wave of the COVID19 pandemic. *Materials and Methods*: This retrospective, single-center observational cohort study included 271 COVID19 patients who received a critical care consult between March 11 and April 30, 2020 during the first wave of the COVID19 pandemic at State University of New York Downstate Health Sciences University. *Results*: Of the 271 patients with critical care consults, 33% survived and 67% expired. At the bivariate level, age, blood urea nitrogen, and blood neutrophil percentage were significantly associated with mortality (mean age: survivors, 61.62 ± 1.50 vs. non-survivors, 68.98 ± 0.85, *p* < 0.001). There was also a significant association between neutrophil% and mortality in the univariate logistic regression model (quartile 4 vs. quartile 1: odd ratio 2.73, 95% confidence interval (1.28–5.82), *p* trend = 0.044). In the multivariate analyses, increasing levels of procalcitonin and C-reactive protein were significantly associated with mortality, adjusting for age, sex, and race/ethnicity (for procalcitonin quartile 4 vs. quartile 1: odds ratio 5.65, 95% confidence interval (2.14–14.9), *p* trend < 0.001). In contrast, higher platelet levels correlated with significantly decreased odds of mortality (quartile 4 vs. quartile 1, odds ratio 0.47, 95% CI (0.22–0.998), *p* trend = 0.010). Of these factors, only elevated procalcitonin levels were associated with intensive care unit acceptance. *Conclusions*: Procalcitonin showed the greatest magnitude of association with both death and likelihood of intensive care unit acceptance at the bivariate level. Our data suggests that procalcitonin reflects pneumonia severity during COVID-19 infection. Thus, it may help the intensivist identify those COVID19 patients who require intensive care unit level care.

## 1. Introduction

As of 15 June 2021, there have been nearly 615,000 COVID-related deaths in the United States, which accounts for the highest number of deaths globally. Of this figure, the state of New York was one of the early epicenters and is the highest contributor with over 53,000 deaths [1]. SUNY Downstate Health Sciences University is located in Kings County, the county with the highest number of COVID deaths in the state and the third highest number of deaths nationwide at almost 11,000 deaths [1]. On 28 March 2020, during the first peak of the COVID-19 outbreak in the United States, SUNY Downstate was designated one of three COVID-only facilities in New York State. 

SUNY Downstate serves a low-income, predominantly Afro-Caribbean community in central Brooklyn. While the disproportionate burden of COVID-19 in the black population has been well-documented [2,3,4,5], there is a paucity of data in the literature focused on the Afro-Caribbean population in the United States. New York City is home to the largest Caribbean contingent in the country and much of this population resides in the central Brooklyn community served by SUNY Downstate. Given the immense burden and continuing impact of COVID-19, ongoing research is needed to uncover and better understand the experience of black and immigrant communities. The current study offers insight from a critical care consult service on COVID-19’s clinical course and outcomes in this population during the first wave of the COVID-19 pandemic.

Studies of critically ill COVID-19 patients identified a state of hyperinflammation characterized by elevated levels of biomarkers including C-reactive protein (CRP), procalcitonin (PCT), and D-dimer [6]. Higher levels of inflammatory markers have been associated with COVID-19 disease severity and death [7]. The use of biomarkers to predict disease severity has proven essential for resource allocation, particularly for respiratory support needs [6]. In this study, we investigated the association between various clinical factors and ICU acceptance and COVID-19 mortality. This study focused on those values available to the intensivist at the time of initial critical care consultation. By doing so, we hoped to identify factors that the critical care team could potentially utilize to better risk stratify and triage COVID-19 patients.

## 2. Materials and Methods

### 2.1. Study Design and Participants

This is a retrospective, single-center, observational study performed at SUNY Downstate Health Sciences University, which was designated by Governor Andrew Cuomo as a COVID-only medical center in New York on 28 March 2020. Patients were identified by querying the electronic medical record (HealthBridge, Eclipsys Sunrise) for critical care initial consult data in patients with a positive COVID-19 test result. The patient visits obtained from the query were cross-checked with a manual log of consults kept by the critical care fellows on service during the time period of interest (28 March 2020 to 30 April 2020). Data on demographics, comorbidities, clinical characteristics, laboratory data within 2 days of admission as well as time of critical care consult, ICU acceptance status, treatment strategies, and outcomes were collected from the electronic medical record according to a set study protocol. Forty-two consults were excluded from the final analysis (11 consults were non-COVID related, 31 were re-consults). A validation step was performed by three independent individuals to ensure that no clerical errors occurred during the compilation of data. After completion of the validation step, we included a total of 271 COVID-19 patients in the final analysis. SUNY Downstate Health Sciences University’s Institutional Review Board approved this study (#1609410-1). They categorized this study as minimal risk research, thereby waiving the informed consent requirement. 

### 2.2. Predictor Variables

Numerous variables were analyzed including age, sex, height, weight, body mass index, vital signs, sodium, potassium, chloride, bicarbonate (HCO^3−^), blood urea nitrogen (BUN), creatinine (Cr), glucose, aspartate aminotransferase (AST), alanine transaminase (ALT), lactate dehydrogenase (LDH), creatine kinase (CK), troponin, brain natriuretic peptide (BNP), lactic acid, C-reactive protein (CRP), procalcitonin, white blood cells, hemoglobin, platelets, neutrophil percentage (PMN%), eosinophil percentage, monocyte percentage, basophil percentage, lymphocyte percentage, prothrombin time (PT), partial thromboplastin time (PTT), international normalized ratio (INR), and D-dimer. These were measured at the time of critical care consultation.

### 2.3. Outcomes

The primary outcome was 60-day in-hospital mortality, and the secondary outcome was ICU acceptance at the time of consult.

### 2.4. Measurement of Covariates

The covariates used in this study were age, sex, and race/ethnicity. These were self-reported by participants at baseline.

### 2.5. Statistical Analysis

Continuous variables were expressed as means and standard deviations and categorical variables were expressed as counts and percentages. Independent sample t-tests were used to assess differences in normally distributed continuous variables based on survival and ICU acceptance status, while Chi-square tests were used to compare categorical variables. No imputation was done for missing data.

Logistic regression models determined whether various risk factors independently predicted the odds of mortality. For each characteristic/risk factor assessed in the logistic regression models, the data was first divided in quartiles (Q1–Q4). Odds ratios were calculated for each quartile, using the first quartile (Q1) as the reference category. The *p*-values for the trend were calculated using the median value of each quartile. 

We created individual multivariable logistic regression models for the following risk factors: procalcitonin, C-reactive protein (CRP), blood urea nitrogen (BUN), platelets, neutrophil percent (PMN%), and lymphocyte percent (lymphocyte%). Models were adjusted for age and sex, and for age, sex, and race/ethnicity. Analyses were performed with GraphPad Prism, and SAS University Edition software (version 9.4 M6, SAS Studio 3.8 interface). A *p*-value of less than 0.05 (two-sided) was considered as statistically significant.

## 3. Results

### 3.1. Patient Demographics

Of the sample of 271 patients with critical care consults, there were 182 non-survivors, representing a 67% mortality rate. Most of the non-survivors were males, *n* = 112 (61.5%), compared to females, *n* = 70 (38.5%) (Table 1). The non-survivors were significantly older (68.98 years ± 0.85) than the survivors (61.62 years ± 1.50), *p* < 0.001. In total, 88% (*n* = 239) of the sample was black, which also mostly comprised the group of non-survivors, *n* = 165 (91%) (Table 1). SOFA scores were calculated on 250 subjects using MDcalc (New York, NY, USA). The SOFA scores (8.33 ± 0.21) of this cohort reflected the high mortality we observed in this study.

### 3.2. Bivariate Analyses

The most common comorbidities in the sample were hypertension (78%) and diabetes (56%), which were similarly distributed in the non-survivors. However, the comorbidities in this study population were not significantly associated with the outcome of mortality. At the bivariate level, all of the laboratory characteristics listed in Table 2 were significantly associated with mortality. A decrease in HCO_3_^−^, lymphocyte percent, and platelet levels were associated with mortality, while an increase in BUN, AST, CRP, PMN%, and procalcitonin were associated with mortality. 

The sample was further stratified by ICU acceptance status (Table 3). Age was found to be associated with acceptance: the mean age of the accepted patients was lower than the unaccepted (65.46 years ± 1.03 vs. 67.99 years ± 0.92), *p* < 0.05. The clinical factors significantly associated with ICU acceptance were HCO_3_^−^, Cr, and procalcitonin.

### 3.3. Multivariate Analyses

We examined several clinical characteristics of interest at the multivariate level to see if they were independently associated with mortality in our study sample. We controlled for age since non-survivors were significantly older than survivors in our sample (Table 1). We also controlled for sex and race/ethnicity since most of the non-survivors were males and African-Americans (Table 1) and blacks have worse COVID-19 health outcomes [2,3,4]. Our multivariate analyses showed a dose–response relationship of the effect of procalcitonin (Table 4 and Figure 1A). Adjusting for age, sex, and race/ethnicity, increasing levels of procalcitonin were significantly associated with the odds of mortality (Q4 vs. Q1, OR 5.65, 95% CI (2.14–14.9), *p* trend < 0.001). C-reactive protein was also significantly associated with mortality at the multivariate level, but mortality did not increase with each quartile as it did with procalcitonin (Figure 1B). The highest odds and most significant association for CRP were seen in the third quartile (Q3 vs. Q1, OR 2.89, 95% CI (1.32–6.32), *p* trend = 0.034) upon controlling for age and sex. BUN showed no dose–response effect (Figure 1C). Adjusting for age, sex, and race/ethnicity, increasing platelet levels across quartiles were significantly associated with decreased odds of mortality (Figure 1D) (Q4 vs. Q1, OR 0.47, 95% CI (0.22–0.998), *p* trend = 0.010). Although increasing neutrophil percent (PMN%) predicted the outcome of mortality in the univariate model (Q4 vs. Q1, OR 2.73, 95% CI (1.28–5.82), *p* trend = 0.044), the trend was not statistically significant after adjusting for age, sex, and race/ethnicity (Figure 1E) (*p* trend = 0.10). As observed with platelets, there was a significant association between increasing lymphocytes and decreased mortality, adjusting for age, sex, and race/ethnicity (Figure 1F) (Q4 vs. Q1, OR 0.55, 95% CI (0.26–1.15), *p*-trend = 0.029).

### 3.4. Assessment of Bacterial Co-Infection

Co-infection with bacteria could account for the increase in procalcitonin identified in our COVID19 subjects. To address this possibility, we examined all respiratory, blood, and urine culture data obtained from the patients 24 h prior and following critical care consultation. We did not identify any cases of bacterial co-infection in this cohort.

## 4. Discussion

This retrospective, single-center, observational study correlated clinical characteristics and laboratory biomarkers with disease severity in COVID-19 patients receiving critical care consultations during the first peak of the pandemic in 2020. Of the 271 unique patients included in the final analyses, the 60-day in-hospital mortality rate was 67%. We attribute the high mortality in our Afro-Caribbean patient population to their disease severity, advanced age, and the limitation in resources available given the large and simultaneous influx of critically ill patients [8]. It is important to note that the black community has a high prevalence of COVID-19 risk factors like diabetes, hypertension, and obesity [2,3,4]. The presence of these cofactors could account for the high mortality in this patient population. Nevertheless, other studies in differing patient populations similarly reported a mortality of 50% or greater in patients in this age cohort who require critical care [9,10] during the first wave of COVID-19 in 2020. Over time, ICU mortality has decreased worldwide [11] and this may be due to new treatments like steroids [12] and Tociluzimab [13] and the fact that the number of severely ill patients no longer exceeds ICU capacity [8]. This current study aimed to identify factors during the pandemic that the intensivist could use at the time of consult to risk stratify COVID-19 inpatients. After performing multivariate analyses, we found that the values of procalcitonin, CRP, platelets, and lymphocyte percentage at the time of consult correlated significantly with mortality. This suggests that the intensivist could use these factors to decide on ICU admission or prognosticate disease outcome in severe COVID-19 inpatients.

Our findings revealed that elevated procalcitonin was most strongly associated with both mortality and ICU acceptance, thereby contributing to the growing body of evidence for the utility of procalcitonin in the context of COVID-19 infection [14,15]. Calcitonin is normally expressed in neuroendocrine cells, but its precursor, procalcitonin (PCT), has been identified as a unique biomarker specific for bacterial sepsis [16]. Serum levels of procalcitonin increase within four hours as part of the innate immune system’s response to an infection [17]. A study performed by Self et al. demonstrated that there is a strong correlation between higher procalcitonin levels and an increased probability of bacterial infection; however, there was no threshold identified that could distinguish bacterial from viral infections [18]. It has been postulated that IFN-ɣ expression in the setting of a viral respiratory tract infection inhibits procalcitonin synthesis [17]. Indeed, studies indicate that viral pneumonia does not elevate procalcitonin levels [19,20] so clinicians can use this marker to identify bacterial lung infections and guide the duration of antibiotic therapy [21]. Procalcitonin levels, however, are frequently elevated in COVID19 and published data suggests that procalcitonin may be a predictor of disease severity in COVID-19. One meta-analysis demonstrated that elevated procalcitonin levels were linked with a 5-fold increased risk of severe COVID-19 infection [22] while another found that elevated procalcitonin, elevated D-dimer, and thrombocytopenia were all associated with severe infection [23]. Similarly, a study from China revealed correlations of IL-6 and procalcitonin levels with COVID-19 severity [14], namely that those with the highest levels of these two biomarkers exhibited significantly increased disease severity. Moreover, a meta-analysis of 25 studies with 5350 patients found that elevated procalcitonin levels were associated with an increased composite poor outcome [RR 3.92 (2.42, 6.35), *p* < 0.001; I^2^: 85%]. Subgroup analyses revealed that elevated procalcitonin was linked to an increased risk of mortality (RR 6.26; I^2^: 96%) and severe COVID-19 [RR 3.93 (2.01, 7.67), *p* < 0.001; I^2^: 63%, *p* = 0.006] [6].

Our analysis of a sample of 271 COVID-19 adult inpatients with critical care consultations found that elevated procalcitonin at the time of consult was strongly associated with mortality and ICU acceptance. The respiratory status of the patient is the primary factor driving ICU admission, so this suggests that procalcitonin elevation reflects the extent of lung injury. Indeed, mortality increased with each increasing quartile of procalcitonin levels and this “dose–response” effect demonstrates the utility of procalcitonin as a disease severity marker in this pandemic. A Louisiana study on hospitalization and mortality among 1382 black and white COVID-positive patients likewise found that elevated levels of procalcitonin were associated with higher in-hospital mortality (HR: 1.40, 95% CI (1.06–1.84)) [2]. In agreement with our study, an early study from China reported that procalcitonin was a marker of disease severity in COVID-positive patients [24]. Similar to our findings in an Afro-Caribbean population, these researchers found that there was a significant increase in the levels of procalcitonin as the disease worsened (PCT levels: 0.05 ± 0.05 ng/mL in the moderate group, 0.23 ± 0.26 ng/mL in the severe group, and 0.44 ± 0.55 ng/mL in the critical group, *p* < 0.05). Together, these findings conducted in a variety of ethnic groups contribute to the growing body of evidence for the utility of procalcitonin in the context of COVID-19 infection. 

Viral infection damages the lung and hinders innate immune responses [25]. This subsequently increases the susceptibility to secondary bacterial infections [26]. During the H1N1 pandemic of 1918, many patients survived influenza viral pneumonia only to subsequently succumb to *S. aureus* and other bacterial lung infections [27]. Early in the pandemic, COVID-19 patients were found to have bacterial co-infection, with one study finding a prevalence of 6.9% [28]. Thus, it is conceivable that bacterial co-infection caused the elevation in procalcitonin in some of the patients observed in this study. To address this possibility, we examined all respiratory, blood, and urine culture data obtained from the patients 24 h prior and following critical care consultation. We did not identify any cases of bacterial co-infection in this cohort. Furthermore, the vast majority of our patients presented during the second week of their illness when they became acutely ill with pulmonary symptoms. Early bacterial co-infection can occur in COVID-19, but it frequently happens days or weeks after viral-mediated lung injury. Moreover, negative respiratory cultures were uncommon in case series of 1918 influenza subjects [27]. The absence of positive respiratory cultures in our cohort provides further evidence against the presence of bacterial pneumonia. Of note, lymphocyte levels were frequently reduced in our patients and this finding is associated with viral rather than bacterial pneumonia [29]. Taken together, these findings indicate that bacterial co-infection did not cause the increases in procalcitonin levels measured in this study. 

Interestingly, we did not observe a strong association between any of the comorbidities and mortality in this patient sample. In fact, only hypertension was significantly, albeit weakly, linked to mortality. Thus, these variables were not further examined in the multivariate analyses. However, an earlier New York study of 257 critically ill patients found that COPD was independently associated with in-hospital mortality (aHR 2·94 1.48–5.84] [30]. Our findings are not unexpected as we did not compare all COVID-19 patients but rather those with greater disease severity who required critical care consultation. Further, it is well documented that comorbidities, such as diabetes and hypertension, are associated with COVID-19 severity in a general patient population [31,32,33]. Beyond these factors, it is important to measure and assess the role of excessive inflammation in response to COVID-19, which this study aimed to do. Research has suggested that elevated levels of biomarkers in black COVID-19 patients may suggest variation in immune response to COVID-19 based on race [2,34]. Nédélec et al., in their research on population differences in immune response to pathogens, concluded that when compared to European ancestry, African ancestry predicts a stronger inflammatory response [34]. This further highlights the importance of evaluating the impact of biomarkers and their role in COVID-19 severity and mortality in our patient population. 

This study focused on procalcitonin since it was the only biomarker that showed a consistent “dose–response” effect and was highly associated with both mortality and ICU acceptance even after adjusting for potential confounders. However, several other markers also correlated with COVID-19 outcomes. Platelets release substances like thrombospondin-1 [35] that counter lung injury and, as reported in other patient groups [36,37], we found that thrombocytopenia correlated with increased mortality in this Afro Caribbean cohort. Similarly, we found that a decreased lymphocyte ratio and an increased neutrophil ratio, and C-reactive protein levels correlated with disease mortality. Again, these findings are consistent with the literature, which linked these parameters to COVID-19 severity and mortality [37,38,39]. The reproducibility of these biomarker findings across several studies conducted in varying patient populations suggests that they could play a useful role in identifying an early time point for those patients with a worse prognosis who may require closer monitoring in a critical care setting.

There are several study limitations that could potentially influence our conclusions. For one, this is a retrospective study. We need further studies of markers like procalcitonin to determine if clinicians could use them prospectively to identify patients who require ICU level care. This is an important clinical question since biomarker scoring systems guide the management of pulmonary embolism [40] and pneumonia [41]. Thus, it is conceivable that medical personnel could use them for triage and prognostication in COVID-19. Secondly, we conducted this study in a largely Afro Caribbean population, so we need additional studies in other patient populations to confirm the generalizability of these findings. Nonetheless, it is encouraging to see that several recent studies found similar conclusions in different ethnic patient communities [36,37]. Given our findings, we expect that future COVID-19 studies will further substantiate procalcitonin as a useful prognostic marker for critical care physicians. Lastly, we conducted this study only on patients undergoing critical care evaluation for possible admission to an intensive care unit. All of these patients had severe respiratory illness requiring high-level oxygen supplementation or some form of mechanical ventilation. For this reason, we cannot extrapolate our findings to those with less severe manifestations in an outpatient or inpatient setting.

## 5. Conclusions

We evaluated demographic and clinical factors associated with ICU acceptance and/or mortality in patients undergoing critical care evaluation during the first wave of the COVID-19 pandemic. Procalcitonin was the one factor strongly associated with both mortality and ICU acceptance in our Afro Caribbean patient population. Moreover, it exhibited a “dose–response” effect, with mortality rising as quartiles of procalcitonin levels increased. These findings indicate that procalcitonin could be used by intensivists to guide COVID-19 management and ICU resource utilization.

## Figures and Tables

**Figure 1 medicina-57-01070-f001:**
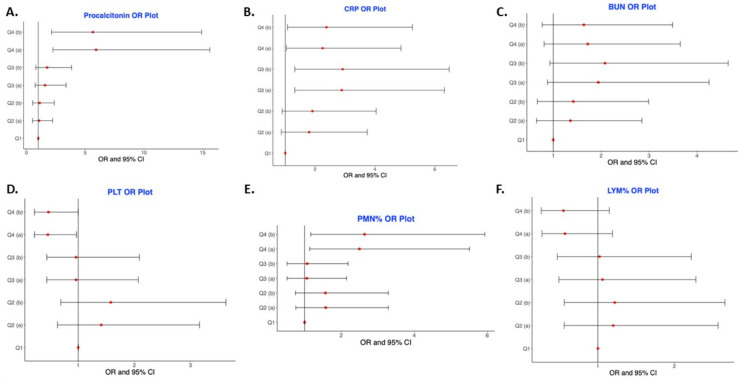
Odds ratio of mortality across quartiles for biomarkers linked with COVID19 mortality. (**A**) Procalcitonin OR Plot; (**B**) CRP OR Plot; (**C**) BUN OR Plot; (**D**) PLT OR Plot; (**E**) PMN% OR Plot; (**F**) LYM% OR Plot.

**Table 1 medicina-57-01070-t001:** Baseline characteristics of critical care consults by survival status.

Characteristics	Survivor *n* (%)	Non-Survivor *n* (%)	Total	*p*-Value
	*n* = 89	*n* = 182	*n* = 271	
Age (mean, SD)	61.62 ± 1.50	68.98 ± 0.85		
Sex				
Female	45 (50.6)	70 (38.5)	115 (42.4)	0.06
Male	44 (49.4)	112 (61.5)	156 (57.6)	
Race/Ethnicity				
Asian	2 (2.25)	0 (0)	2 (0.74)	0.10
Black	74 (83.2)	165 (90.7)	239 (88.2)	
Hispanic	4 (4.49)	2 (1.10)	6 (2.21)	
White	4 (4.49)	7 (3.85)	11 (4.06)	
Unknown	5 (5.62)	8 (4.40)	13 (4.80)	
Comorbidities				
Asthma	8 (9)	17 (9.34)	25 (9.23)	0.93
COPD	6 (6.74)	16 (8.79)	22 (8.12)	0.56
Diabetes	48 (53.9)	104 (57.1)	152 (56.1)	0.62
Hypertension	63 (70.8)	148 (81.3)	211 (77.9)	0.05
HIV	1 (1.1)	7 (3.85)	8 (2.95)	0.21
CKD	12 (13.5)	29 (15.9)	41 (15.1)	0.60

**Table 2 medicina-57-01070-t002:** Age and laboratory findings of critical care consults by survival status.

Factors	Survivor		Non-Survivor		*p*-Value
	*n*	Mean (SD)	*n*	Mean (SD)	
Age	89	61.62 ± 1.50	182	68.98 ± 0.85	<0.0001
HCO_3_^−^	88	21.95 ± 0.76	181	20.66 ± 0.45	<0.02
BUN	88	36.02 ± 3.27	182	47.87 ± 3.34	<0.004
Cr	88	2.43 ± 0.30	182	2.91 ± 0.24	<0.02
AST	85	82.44 ± 16.43	165	241.4 ± 106.6	<0.02
LDH	47	636.3 ± 84.10	88	740 ± 55.14	<0.05
CRP	49	161.7 ± 16.04	94	230.4 ± 12.56	<0.002
PLT	85	292.2 ± 14.82	179	227.6 ± 8.36	<0.0001
Neutrophil %	73	79.98 ± 1.12	155	83.57 ± 0.63	<0.004
Lymphocyte %	73	11.81 ± 0.97	154	9.06 ± 0.45	<0.009
PT	46	13.83 ± 0.32	69	18.13 ± 2.11	<0.05
Procalcitonin	47	3.155 ± 2.02	77	10.18 ± 3.01	<0.0001
Troponin	44	0.40 ± 2.02	84	2.60 ± 2.02	<0.03

**Table 3 medicina-57-01070-t003:** Age and laboratory findings of critical care consults by ICU acceptance.

Factors	Not Accepted		Accepted		*p*-Value
	*n*	Mean (SD)	*n*	Mean (SD)	
Age	115	67.99 ± 0.92	156	65.46 ± 1.03	<0.05
HCO_3_^−^	113	21.88 ± 0.44	156	20.39 ± 0.53	<0.03
BUN	114	42.56 ± 3.24	156	44.96 ± 3.06	0.32
Cr	114	2.35 ± 0.22	156	2.91 ± 0.24	<0.03
AST	102	300.0 ± 148.3	148	134.4 ± 33.60	0.09
LDH	50	647.5 ± 52.37	85	758.5 ± 61.41	0.17
CRP	58	212.4 ± 12.61	85	210.7 ± 13.96	0.10
PLT	110	252.4 ± 9.39	154	241.1 ± 9.53	0.37
PMN%	94	82.18 ± 1.13	134	82.53 ± 0.69	0.33
Lymphocyte%	94	9.09 ± 0.54	133	1.00 ± 0.56	0.10
PT	39	17.72 ± 2.39	76	15.07 ± 0.69	0.42
Procalcitonin	45	2.84 ± 0.67	79	10.50 ± 3.13	<0.009
Troponin	41	0.56 ± 0.17	87	2.50 ± 1.72	0.24

**Table 4 medicina-57-01070-t004:** Logistic regression models of mortality prediction (with *p*-value for trend).

Characteristic	Q1 (Referent)	Q2	Q3	Q4	*p*-Trend
	OR	OR (95% CI)	OR (95% CI)	OR (95% CI)	
Procalcitonin					
Model ^a^	1.00	1.07 (0.51–2.21)	1.58 (0.74–3.37)	5.92 (2.25–15.6)	<0.001
Model ^b^	1.00	1.12 (0.53–2.36)	1.75 (0.79–3.84)	5.65 (2.14–14.9)	<0.001
*CRP*					
Model ^a^	1.00	1.80 (0.87–3.74)	2.89 (1.32–6.32)	2.25 (1.04–4.87)	0.034
Model ^b^	1.00	1.91 (0.90–4.04)	2.92 (1.32–6.48)	2.38 (1.08–5.25)	0.049
*BUN*					
Model ^a^	1.00	1.36 (0.65–2.85)	1.94 (0.88–4.25)	1.72 (0.81–3.65)	0.22
Model ^b^	1.00	1.42 (0.67–2.99)	2.08 (0.93–4.65)	1.64 (0.77–3.49)	0.31
*PLT*					
Model ^a^	1.00	1.41 (0.63–3.16)	0.96 (0.44–2.07)	0.46 (0.22–0.97)	0.010
Model ^b^	1.00	1.58 (0.69–3.63)	0.96 (0.44–2.09)	0.47 (0.22–0.998)	0.010
PMN% (neutrophils)					
Model ^a^	1.00	1.58 (0.76–3.29)	1.06 (0.52–2.15)	2.50 (1.14–5.51)	0.12
Model ^b^	1.00	1.57 (0.75–3.29)	1.07 (0.52–2.19)	2.64 (1.17–5.93)	0.10
Lym%					
Model ^a^	1.00	1.20 (0.56–2.57)	1.06 (0.49–2.28)	0.57 (0.27–1.19)	0.039
Model ^b^	1.00	1.22 (0.56–2.66)	1.02 (0.47–2.22)	0.55 (0.26–1.15)	0.029

^a^ Adjusted for Age and Sex ^b^ Adjusted for Age, Sex, and Race/Ethnicity.

## Data Availability

Data is contained within the article.

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
