# Peer review of "Procalcitonin Levels in COVID-19 Patients Are Strongly Associated with Mortality and ICU Acceptance in an Underserved, Inner City Population"

_medicina, 2021, doi:10.3390/medicina57101070_

Round 1
Reviewer 1 Report
The draft "Procalcitonin Levels in COVID-19 Patients are Strongly Associated with Mortality and ICU Acceptance in an Underserved, Inner City Population" aimed to identify demographic and clinical factors at the time of critical care consultation that were associated with mortality or ICU acceptance in a predominantly Afro-Caribbean population during the first wave of the COVID19 pandemic. The retrospective, single-center observational cohort study included 271 COVID19 patients who received a critical care consult. The authors conclude, that Procalcitonin was the factor with the greatest magnitude of association with both death and the likelihood of ICU acceptance at the bivariate level and it therefore it may help the intensivist identify those COVID19 patients who require ICU level care.
Despite various publications on the subject, the topic is still of interest, especially against the background of an efficient allocation of limited resources. The draft is written in an understandable and comprehensible way and the statistical methodology seems adequate.
Nevertheless, some points should be re-evaluated and/or presented in more detail:
- The entire draft should be more focused on the actual parameters investigated (epidemiology/laboratory results).
- As mentioned by the authors, data already exist on African-American populations. How is the distinction from the current study population made?
- Why were only data from a short period of the first wave used? By now, data from the following waves should also be available. Can the trends described be confirmed in these as well?
- Key wording/assumptions should be defined more precisely. For example, why is the study population referred to as "underserved"?
- How was the decision to admit to the ICU initially made? Was the PCT-level already taken into account here? If so, this would be a strong confounder with regard to the prediction for ICU admission.
- The presentation of results should be limited to the multivariate adjusted numbers to avoid confusion. Furthermore, all results should be presented (currently, some new results can be found in the discussion, especially in the 4th paragraph).
- Figures of the ORs would help to provide a quicker overview.
- General formal requirements should be adhered to (e.g., writing out abbreviations in the abstract, correct headings, etc.).
Author Response
We thank the reviewer for their insightful comments. We incorporated their suggestions to improve the revised manuscript. Below, is a point-by-point response to the important critiques raised by the reviewer.
REVIEWER 1 COMMENTS:
- The entire draft should be more focused on the actual parameters investigated (epidemiology/laboratory results).
Response: The manuscript primarily focuses on the role of procalcitonin in this cohort since this marker best predicted mortality and likelihood of ICU acceptance.
- As mentioned by the authors, data already exist on African- American populations. How is the distinction from the current study population made?
Response: The main findings from our study were that procalcitonin and thrombocytopenia were prognostic markers in this cohort. There are numerous studies linking these parameters in a general patient population but few to none in an African American cohort. Searching the literature using the terms “African Americans”, “SARS-CoV-2” and “thrombocytopenia” yielded no results. Replacing the term “thrombocytopenia” with “procalcitonin” produced five results. A unique feature of our patient population is that it has a large Afro-Caribbean representation and this community is not well studied in the literature. Further, not much is documented on the clinical course and outcomes of an African-American (specifically Afro-Caribbean) patient population, from the perspective of a critical care consult service, especially at a time when New York City was the epicenter of the Covid-19 disease.
- Why were only data from a short period of the first wave used? By now, data from the following waves should also be available. Can the trends described be confirmed in these as well?
Response: One of the main goals of this study was to examine retrospectively what factors correlated with worse outcomes and likelihood of ICU admission in patients undergoing critical care evaluation during this crisis period. The purpose was to identify prognostic markers that could potentially be prospectively used for triage purposes in the future. As a result of these studies, our critical care group became aware of the prognostic importance of procalcitonin. Thus, examining subsequent waves would confound our data as this knowledge would potentially influence the decision to admit to an ICU.
- Key wording/assumptions should be defined more precisely. For example, why is the study population referred to as "underserved"?
Response: Many of our immigrant Afro Caribbean patients do not seek out routine medical care before they develop a health crisis requiring hospitalization. For this reason, the medical community categorizes these patients as “underserved”. We replaced the word “underserved” with the term “vulnerable” which would be more apt.
- How was the decision to admit to the ICU initially made? Was the PCT-level already taken into account here? If so, this would be a strong confounder with regard to the prediction for ICU admission.
Response: Initially, our critical care group was not aware of any data documenting procalcitonin levels as a reliable marker of disease severity and prognosis. Thus, procalcitonin levels were unlikely to be a deciding factor in the decision to admit to the ICU during the first wave. After the first wave, our intensivists were aware of the prognostic value of this marker and this could confound analyses of subsequent COVID-19 waves we experienced.
- The presentation of results should be limited to the multivariate adjusted numbers to avoid confusion. Furthermore, all results should be presented (currently, some new results can be found in the discussion, especially in the 4th paragraph).
Response: We have updated Table 4 to show only the multivariate adjusted numbers. We now include in the results a section on the assessment of bacterial co-infection.
- Figures of the ORs would help to provide a quicker overview.
Response: We have included OR plots in the revised manuscript that correspond with the regression analyses. These are provided in Figure 1.
- General formal requirements should be adhered to (e.g., writing out abbreviations in the abstract, correct headings, etc.)
Response: We removed the abbreviations from the abstract and corrected headings in the manuscript.
Reviewer 2 Report
This interesting paper correlates PCT and Covid mortality. I think that some data should be added to improve patients description: at least, the SOFA score and the P/F ratio, ventilation mode
Author Response
We thank the reviewer for their insightful comments. We incorporated their suggestions to improve the revised manuscript. Below, is our response to this important critique raised by the reviewer.
REVIEWER 2 COMMENTS:
- This interesting paper correlates PCT and Covid mortality. I think that some data should be added to improve patients’ description: at least, the SOFA score and the P/F ratio, ventilation mode.
Response: We now provide SOFA score information for our cohort on pages 6 and 7 of the revised manuscript. The high SOFA reflects the disease severity of this group.